# Prostate Artery Embolization (PAE) in the Treatment of Benign Prostatic Hyperplasia: A Case Series and Narrative Review

**DOI:** 10.3390/jcm14113775

**Published:** 2025-05-28

**Authors:** Vincenzo Iossa, Ernesto Punzi, Savio Domenico Pandolfo, Gianluca Spena, Pierluigi Russo, Carlo Giulioni, Achille Aveta, Lorenzo Spirito, Giulio Lombardi, Vittorio Imperatore

**Affiliations:** 1Department of Urology, Azienda Ospedaliera “S.G. Moscati”, 83100 Avellino, Italy; vittorio.imperatore@aornmoscati.it; 2Department of Radiology, Azienda Ospedaliera “S.G. Moscati”, 83100 Avellino, Italy; ernesto.punzi@aornmoscati.it (E.P.); giulio.lombardi@aornmoscati.it (G.L.); 3Department of Urology, University of L’Aquila, 67100 L’Aquila, Italy; pandolfosavio@gmail.com; 4Department of Neurosciences, Reproductive Sciences and Odontostomatology, “Federico II” University, 80138 Naples, Italy; 5Department of Urology, Istituto Nazionale Tumori, IRCCS, “Fondazione G. Pascale”, 80131 Naples, Italy; spena.dr@gmail.com (G.S.); achille-aveta@hotmail.it (A.A.); 6Department of Urology, Fondazione Policlinico Universitario Agostino Gemelli, IRCCS, 00168 Rome, Italy; pierluigi.russo01@icatt.it; 7Urology Unit, Casa di Cura Villa Igea, 60127 Ancona, Italy; carlo.giulioni9@gmail.com; 8Unit of Urology, Department of Woman, Child and General and Specialized Surgery, University of Campania “Luigi Vanvitelli”, 80131 Naples, Italy; lorenzospirito@msn.com

**Keywords:** prostatic artery embolization, benign prostatic hyperplasia, lower urinary tract symptoms

## Abstract

**Background/Objectives:** Prostatic artery embolization (PAE) has emerged as a minimally invasive alternative for treating lower urinary tract symptoms (LUTS) secondary to benign prostatic hyperplasia (BPH), particularly in high-risk surgical candidates. This study aims to evaluate the efficacy, safety, and clinical outcomes of PAE, combining a retrospective case series with a narrative review of the literature. **Methods**: A single-center retrospective analysis was conducted on 10 patients aged ≥ 70 years with moderate-to-severe LUTS due to BPH who underwent PAE between January 2021 and January 2024. Inclusion criteria included IPSS > 18, Qmax < 12 mL/s, prostate volume > 45 cc, and resistance to medical therapy. Embolization was performed using 300–500 µm tris-acryl gelatin microspheres via the PErFecTED technique. Follow-up included IPSS, Qmax, prostate volume (PV), PSA levels, and complications. A narrative review of 18 studies (*n* = 1539 patients) was also conducted to contextualize findings. **Results**: Technical success was achieved in all patients (100%), and clinical success (IPSS reduction ≥ 50%) in 90%. At 12 months, the following significant improvements were observed: mean IPSS decreased from 24 to 12 (*p* < 0.0001), Qmax increased from 8.7 to 12.6 mL/s (*p* < 0.0001), PV reduced from 66.4 to 49.4 cc (*p* < 0.0001), and PSA from 5.0 to 3.4 ng/mL (*p* < 0.0001). Outcomes remained stable up to 36 months. Two patients developed transient post-procedural fever; no major complications were recorded. **Conclusions**: PAE is a safe and effective treatment for LUTS related to BPH, offering durable symptom relief and minimal morbidity, particularly in elderly and comorbid patients. While the evidence supports its role as an alternative to TURP, larger prospective trials are necessary to confirm its long-term efficacy and optimize patient selection.

## 1. Introduction

Prostatic artery embolization (PAE) has emerged as a minimally invasive alternative for the management of lower urinary tract symptoms (LUTS) secondary to benign prostatic hyperplasia (BPH) [1]. This endovascular procedure involves the selective embolization of the prostatic arteries, resulting in ischemic shrinkage of the gland and subsequent symptom relief [2]. PAE offers the advantage of being performed under local anesthesia, with reduced hospitalization time and a lower risk of sexual dysfunction compared with traditional surgical approaches [3].

While initial studies demonstrated promising outcomes in terms of symptom improvement and quality of life, conflicting evidence remains regarding its long-term efficacy and safety compared with transurethral resection of the prostate (TURP), the current gold standard [4]. Despite this, its higher risk of bleeding, dilutional hyponatremia, and ejaculatory dysfunction underscores the need for alternative approaches, particularly in high-risk patients [5,6]. In this context, an increasing body of the literature highlights PAE as an effective therapeutic option, particularly in elderly patients with multiple comorbidities and those on anticoagulation therapy, offering symptomatic relief without the need to interrupt antithrombotic treatment [7,8]. Compared with other minimally invasive procedures, such as Holmium laser enucleation of the prostate, photoselective vaporization, and transurethral water vapor therapy, PAE stands out for its safety profile and preservation of sexual function [9,10,11,12,13,14].

Despite the growing international evidence supporting PAE, including six randomized controlled trials and multiple systematic reviews, real-world data from low-volume centers remain scarce [15]. Real-world evidence from such contexts—together with detailed technical descriptions—may offer valuable insights for centers approaching PAE implementation. In this study, we primarily aim to describe the procedural technique adopted in our institution and report the initial clinical outcomes of a small case series, representing the first experience with PAE in our center. Additionally, we conducted a comprehensive review of the existing literature to contextualize our findings and support the clinical rationale for PAE in the treatment of BPH-related LUTS.

## 2. Materials and Methods

### 2.1. Study Design

This is a retrospective single-center cohort study designed to evaluate the safety, efficacy, and outcomes of PAE in patients with BPH and moderate-to-severe LUTS. The study adhered to the guidelines outlined by the Declaration of Helsinki and received approval from the institutional review board. Written informed consent was obtained from all participants prior to the procedure.

### 2.2. Patient Selection

Between January 2021 and January 2024, patients were recruited from the outpatient urology and interventional radiology clinics. Inclusion and exclusion criteria were pre-defined to ensure patient safety and homogeneity.

All patients underwent a comprehensive preoperative evaluation, which included a detailed medical history and physical examination. The severity of LUTS was assessed using the international prostate symptom score (IPSS). Prostate volume was measured via transrectal ultrasound (TRUS), while serum prostate-specific antigen (PSA) levels and maximum urinary flow rate (Qmax) were also evaluated. Additionally, post-void residual (PVR) urine volume was assessed to further characterize baseline urinary function.

#### 2.2.1. Inclusion Criteria

Age ≥ 70 years;International prostate symptom score (IPSS) > 18;Maximum urinary flow rate (Qmax) < 12 mL/s;Prostate volume > 45 cc, measured by transrectal ultrasound (TRUS);Resistance to medical therapy for ≥ 6 months;Acceptance of potential risks, including post-treatment sexual dysfunction.

#### 2.2.2. Exclusion Criteria

Prostate cancer diagnosis or suspicion;Previous prostate surgery;Severe renal impairment (eGFR < 30 mL/min/1.73 m^2^);Active urinary tract infection;Contraindications to embolization, such as severe atherosclerosis.

### 2.3. Embolization Procedure

#### 2.3.1. Arterial Access and Imaging

Arterial access was achieved through the right radial or femoral artery using a 5 Fr introducer sheath. Once access was secured, a 5 Fr catheter was advanced into the internal iliac artery on the ipsilateral side to facilitate targeted embolization. To visualize the arterial anatomy and identify the prostatic arteries, pelvic angiography and cone beam computed tomography (CBCT) were performed using a power injector, delivering 20 mL of contrast medium at a rate of 4 mL/s with 900 psi pressure (Figure 1).

#### 2.3.2. Catheterization and Embolization

Following arterial imaging, a 1.9–2.4 Fr microcatheter was advanced into the prostatic artery and carefully positioned distally to avoid branches supplying the urethral and capsular zones. Any dangerous anastomoses, such as those connecting to rectal or penile arteries, were either embolized using coils or meticulously avoided (Figure 2). Embolization was performed using 300–500 µm tris-acryl gelatin microspheres, diluted with contrast medium and saline, to achieve effective arterial occlusion. The PErFecTED approach (proximal embolization first, then embolize distal) was employed to enhance ischemic efficacy while minimizing recurrence rates [16]. Bilateral embolization was completed in 8 patients, while 2 underwent unilateral PAE due to vascular calcifications or the presence of an aortobifemoral bypass.

#### 2.3.3. Post-Procedure Imaging and Verification

Upon completion of the embolization, a final angiogram was performed to confirm the occlusion of the prostatic arteries. Additionally, a non-contrast CBCT scan was conducted to verify the uniform distribution of embolic material within the prostate parenchyma, ensuring the procedure’s technical success (Figure 3) [17].

#### 2.3.4. Postoperative Care and Follow-Up

Patients were monitored for 24–48 h post-procedure for early complications and discharged once stable. Follow-up assessments were conducted at 1, 3, and 6 months, then every 6 months and annually thereafter. Each follow-up visit included IPSS scoring; prostate volume measurement by TRUS; serum PSA levels; and Qmax.

The primary endpoint was the mean change in IPSS at 12 months post-procedure. Secondary endpoints included changes in prostate volume, PSA levels, Qmax, and adverse events.

### 2.4. Statistical Analysis

Data were analyzed using GraphPad Prism (version 9.0, GraphPad Software, San Diego, CA, USA). Continuous variables were expressed as means with standard deviation (SD) for normally distributed data, or as medians with interquartile ranges (IQRs) when the data were not normally distributed. Categorical variables were presented as frequencies and percentages to facilitate comparison across groups. Paired *t*-tests were used to compare pre- and post-procedure continuous variables when the data followed a normal distribution, while Wilcoxon signed-rank tests were applied for non-normally distributed variables. Chi-square or Fisher’s exact tests were employed for categorical variables, depending on sample size and expected frequencies. A *p*-value of less than 0.05 was considered statistically significant. Interobserver agreement for imaging and clinical evaluations was assessed using Cohen’s kappa coefficient, providing a measure of consistency between evaluators. Missing data were managed using a pairwise deletion approach, ensuring that analyses were conducted with all available data while minimizing potential biases.

## 3. Results

This study included 10 consecutive patients with BPH and moderate-to-severe LUTS who underwent PAE. The mean age of the cohort was 79.5 years (range: 74–87 years). Bilateral embolization was performed in eight patients (80%), while two patients (20%) underwent unilateral PAE due to vascular calcifications or the presence of an aortobifemoral bypass. The mean procedural duration was 95 min, and the average length of hospital stay was 2.5 days.

Technical success, defined as successful embolization of the prostatic artery on at least one side, was achieved in all cases (100%). Clinical success, defined as an IPSS reduction of ≥50% without the need for additional treatment, was observed in 9 out of 10 patients (90%).

At the 12-month follow-up, statistically significant improvements were observed across all evaluated parameters. The mean IPSS decreased from 24 to 12 points, representing an average reduction of 12 points (*p* < 0.0001). The mean Qmax increased from 8.7 to 12.6 mL/s (*p* < 0.0001). PV showed a significant reduction, decreasing from a mean of 66.4 to 49.4 cc (*p* < 0.0001). Similarly, PSA levels declined from a mean of 5.0 to 3.4 ng/mL (*p* < 0.0001). These results are summarized in Table 1.

No significant differences were noted between 12-month and subsequent follow-up assessments, indicating stable outcomes over time. Patients were monitored for a median duration of 24 months (range: 12–36 months). Regarding safety, two patients (20%) developed transient fever (T > 38 °C), which was resolved with NSAIDs. No major complications, including bladder ischemia or persistent perineal pain, were reported during the follow-up period.

## 4. Discussion and Literature Context

The present study represents the initial clinical experience of our center with PAE. Its primary aim is to describe the technical aspects of the procedure and provide preliminary real-world data on safety and functional outcomes in a case series of patients. This descriptive approach is intended to offer practical insights for other institutions considering the adoption of PAE in clinical practice. PAE is increasingly recognized as a promising minimally invasive therapeutic option for managing LUTS associated with BPH [3,4,5,7]. While TURP remains the gold standard for surgical treatment, PAE has emerged as a valuable alternative, particularly in patients at high surgical risk or those on anticoagulation therapy [18].

The therapeutic efficacy of PAE is based on two principal mechanisms. First, extensive devascularization of the enlarged prostate induces ischemia, necrosis, and apoptosis, leading to significant prostate shrinkage. This may also enhance androgen-related apoptosis by disrupting the local androgen supply. Second, PAE appears to impair prostatic innervation, reducing smooth muscle tone through a decreased density of α1-adrenergic receptors, thereby improving urinary flow by lowering urethral resistance [3,19,20].

Advanced techniques, such as the PErFecTED approach used in our study, optimize embolization outcomes by targeting both proximal and distal arteries, which helps reduce recurrence rates and complications. Although larger cohorts are needed for definitive validation, this technique has demonstrated superior ischemic efficacy and sustained symptom relief compared with conventional PAE [16].

Various embolic agents have been employed in clinical trials of PAE for the treatment of LUTS secondary to BPH, including non-spherical polyvinyl alcohol (PVA) particles (Cook Medical, Bloomington, IN, USA), tris-acryl gelatin microspheres (Embosphere^®^, Merit Medical, Singapore), and Embozene^®^ Microspheres (CeloNova BioSciences, San Antonio, TX, USA). Unlike traditional PVA particles, which tend to aggregate and may result in proximal embolization, microspheres, owing to their uniform size and compressibility, allow for deeper intraprostatic penetration and more effective ischemia [21,22].

In our comprehensive review, 18 studies comprising 1539 patients were analyzed, demonstrating consistent outcomes across all evaluated parameters [7,23,24,25,26,27,28,29,30,31,32,33,34,35,36,37,38,39,40] (Table 2).

Carnevale et al. were the first to report, in 2008, the use of prostatic artery embolization as a therapeutic approach for BPH in humans, successfully restoring spontaneous voiding in two patients previously dependent on urinary catheters [23]. Since then, numerous non-randomized studies across various countries have confirmed the procedure’s favorable safety profile and efficacy.

The most extensive and long-term study on PAE was conducted by Pisco et al., published in 2016 [7]. This investigation included 630 men affected by BPH who presented with moderate-to-severe LUTS, either unresponsive to medical treatment for at least six months or unwilling to pursue pharmacological therapy. Bilateral embolization was successfully achieved in 572 patients (92.6%), while 46 patients (7.4%) underwent unilateral embolization. The cumulative rates of clinical success were 81.9% (95% confidence interval [CI], 78.3–84.9%) at mid-term and 76.3% (95% CI, 68.6–82.4%) at long-term follow-up with statistically significant improvements (*p* < 0.0001) in IPSS, quality of life, prostate volume, PSA, maximum urinary flow rate, post-void residual, and erectile function (IIEF). Only two major complications were reported, both of which resolved without lasting consequences [7].

PAE has shown particularly encouraging results in patients with very large prostates (>90 cm^3^), often accompanied by severe LUTS. Three prospective studies have demonstrated substantial clinical benefit in this population, with mean reductions in IPSS ranging from 12 to 15 points and corresponding improvements in quality-of-life scores by 2 to 4 points, in the absence of major adverse events. Additional comparative analyses assessing outcomes in large versus medium-sized prostates further reinforced the safety and efficacy of PAE in larger glands. Remarkably, one study reported even greater symptom relief in patients with larger prostates, with successful embolization achieved in volumes up to 274 cm^3^ [25,26,27,28,29,30].

To date, only four randomized controlled trials (RCTs) have compared PAE with TURP. While TURP demonstrates superior efficacy in rapidly reducing prostate volume and alleviating obstruction, it is associated with increased rates of bleeding, retrograde ejaculation, and extended recovery periods [41]. Conversely, PAE provides comparable symptom relief with fewer complications, particularly in high-risk populations [7,42,43].

The first RCT by Gao et al. in 2014 reported similar improvements in voiding symptoms between the two groups, although PAE had higher technical and clinical failure rates [35]. TURP, on the other hand, was associated with greater perioperative morbidity, including bleeding and TUR syndrome. Notably, PAE resulted in shorter hospital stays and less catheterization [35]. A more recent single-center RCT by Carnevale et al. introduced a third arm using the PErFecTED technique [16,44]. Although TURP and PErFecTED PAE both demonstrated greater symptom relief compared with conventional PAE, no significant difference in IPSS was observed between TURP and PErFecTED. Nevertheless, the enhanced functional outcomes observed with TURP were accompanied by increased adverse events and prolonged hospitalization [16,44]. These findings underscore the potential of PAE as a viable alternative, especially for elderly patients or those on anticoagulant therapy.

Only one RCT has incorporated urodynamic studies post-PAE, revealing that, at three months post-intervention, only one-third of patients exhibited a non-obstructive urodynamic profile, while the remaining two-thirds displayed equivocal or obstructive results [40]. IPSS reduction following PAE has shown variability across studies, ranging from 9.2 to 21.0 points [38,40].

Our results are consistent with previous studies demonstrating marked improvements in LUTS following PAE. At the 12-month follow-up, the mean IPSS reduction of 12 points, the Qmax increase of 3.9 mL/s, the PV decrease of 17.0 cc, and the PSA reduction of 1.6 ng/mL reflect the effectiveness of this approach. In this regard, uroflowmetry remains a critical component for objectively assessing functional outcomes post-PAE and should be routinely integrated into follow-up in a patient-friendly manner [45].

Despite the rare risk of accidental embolization in the pelvic region, which can lead to severe complications such as bladder necrosis or rectal ischemia in centers with limited experience, prostatic artery embolization (PAE) is confirmed to be a procedure with an overall favorable safety profile [46]. In our study, only two patients (20%) experienced transient fever, resolved with NSAIDs, and no major complications were reported. This aligns with previous large-scale studies, such as those by Pisco et al. and Wang et al., which reported major complication rates of 0.3% and 0.4%, respectively [7,47]. Minor complications like dysuria and post-embolization syndrome (PES) were self-limiting, typically resolving within days. Manifestations of PES, encompassing nausea, vomiting, and fever, alongside a transient exacerbation of pre-existing symptoms, particularly irritative voiding symptoms due to an increased post-embolic inflammatory process, should be categorized as anticipated side effects rather than complications. While the intensity of PES remains unpredictable and typically resolves within a week, effective management is essential for expedited recovery, improved patient satisfaction, and enhanced quality of life [3].

Gao et al. reported a 25.9% incidence of early acute urinary retention (AUR) within the first month following prostatic artery embolization (PAE) [35]. Furthermore, the same study reported six cases of post-embolization syndrome (PES) within the first 30 days post-PAE. However, these complications were not explored comprehensively, and patients affected by AUR required temporary urinary catheterization and pharmacotherapy [35].

Instead, in a cohort of 81 patients, Insausti et al. documented a 13.6% incidence of minor complications, totaling 13 events [39]. The specific complications reported included urinary tract infections (27.3%), urinary retention (27.3%), post-embolization syndrome (27.3%), erectile dysfunction (9%), and femoral artery dissection (9%). Management strategies varied as follows: patients with urinary retention underwent TURP, those with urinary tract infections received medical treatment, and the remaining complications resolved spontaneously. All patients achieved recovery without lasting sequelae [39].

Despite promising outcomes, several limitations warrant consideration. The heterogeneity of patient populations, differences in embolization techniques, and variations in inclusion criteria across studies introduce potential biases. Additionally, long-term data remain limited, with most studies reporting outcomes up to 12 or 24 months. Further prospective, multicenter, and randomized controlled trials are essential to establish the durability of symptom relief and clarify indications for PAE in specific patient subgroups, such as those with large prostate volumes.

This study presents several limitations that should be acknowledged. First, the retrospective and single-center nature of the investigation may introduce selection and reporting biases, limiting the generalizability of our findings. Second, the small sample size of only ten patients restricts the statistical power of the analysis and may not capture the full spectrum of potential complications or outcomes associated with PAE. Third, although follow-up extended up to 36 months for some patients, the median follow-up duration was 24 months, which may be insufficient to fully assess long-term durability, especially when compared with standard surgical approaches. Additionally, the lack of a control group or direct comparison with other minimally invasive techniques such as HoLEP or Rezum limits the strength of our conclusions regarding the relative efficacy of PAE. Finally, while procedural standardization was attempted, operator experience and anatomical variability may have influenced technical success and outcomes, particularly in cases requiring unilateral embolization due to vascular challenges.

Nonetheless, one of the strengths of this study lies in the detailed description of the embolization technique, which may serve as a valuable reference for clinicians aiming to implement or refine PAE in their practice.

## 5. Conclusions

This case series and comprehensive review confirm that PAE is a safe and potentially effective minimally invasive option for managing LUTS in patients with BPH, particularly those at high surgical risk. Beyond reporting preliminary clinical outcomes, the technical description of the PAE procedure may serve as a practical reference for centers approaching PAE for the first time. Despite the small sample size, the observed improvements in IPSS, Qmax, PV, and PSA—combined with the absence of major complications—support the feasibility and safety of the technique. Further prospective multicenter studies are warranted to validate these findings and clarify the role of PAE in the broader therapeutic landscape of BPH.

## Figures and Tables

**Figure 1 jcm-14-03775-f001:**
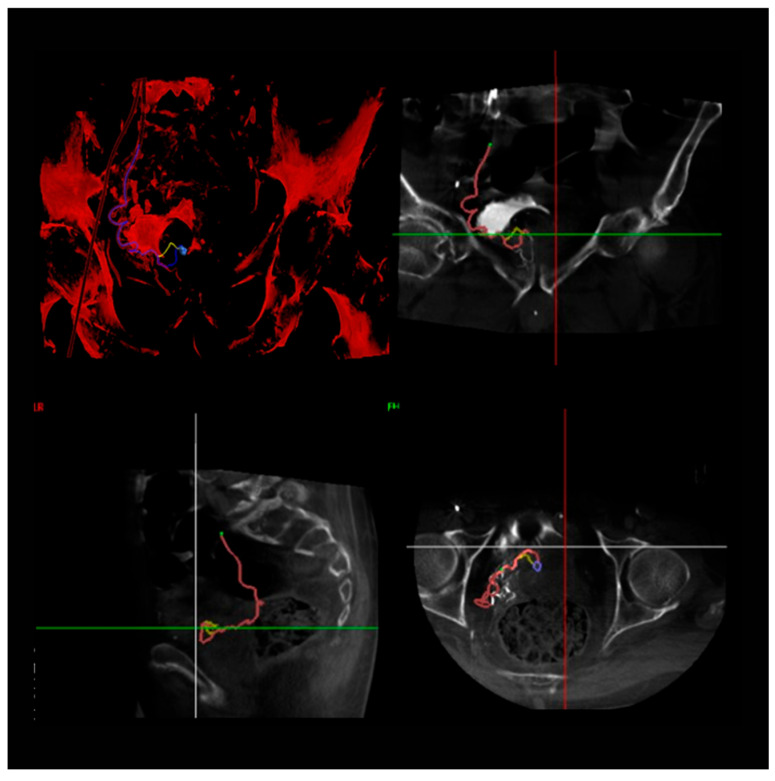
The figure illustrates the use of a dedicated software tool (EmboGuide, Philips—R1.4) during prostatic artery embolization. Following the acquisition of a cone beam computed tomography (CBCT) scan with contrast medium injected into the internal iliac artery on one side, the software identifies the arterial branch supplying the ipsilateral prostatic lobe. Additionally, it enables the fusion of three-dimensional CBCT images with real-time two-dimensional fluoroscopy, thereby reducing procedural time, radiation dose, and the volume of contrast medium administered to the patient.

**Figure 2 jcm-14-03775-f002:**
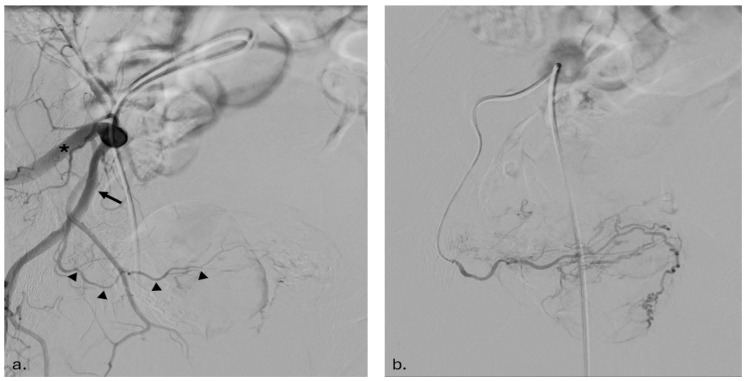
Right internal iliac artery angiography (**a**,**b**). Selective angiography of the right internal iliac artery in an ipsilateral oblique projection. This view allows separation of the anterior division trunk (arrow), which typically gives rise to the prostatic artery (arrowhead), from the posterior division trunk (asterisk), which gives rise to parietal muscular branches (**a**). Super-selective angiography of the ipsilateral prostatic artery performed using a 1.9 Fr catheter and an injector (10 mL of contrast medium at a flow rate of 2 mL/s; 600 psi). This detailed imaging is crucial for targeted embolization, as it allows identification of potential anastomoses with non-target vessels, such as rectal or penile arteries (**b**).

**Figure 3 jcm-14-03775-f003:**
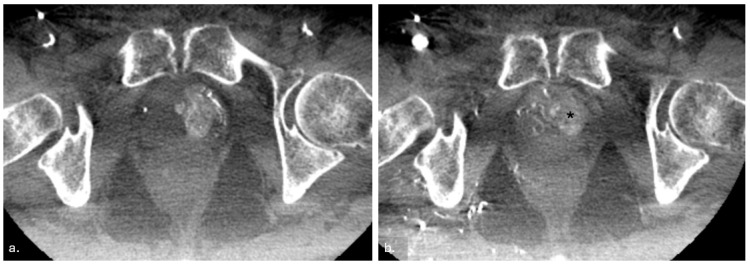
This case involves a patient presenting with hematuria and a mildly enlarged prostate volume (**a**,**b**). Cone beam CT (CBCT) obtained after contrast medium injection into the LEFT internal iliac artery. The image demonstrates contrast enhancement of the ipsilateral prostatic parenchyma, confirming vascular supply to the prostate on the left side (**a**). CBCT obtained after contrast medium injection into the RIGHT internal iliac artery. The image highlights contrast enhancement of the ipsilateral prostatic lobe and reveals the distribution of hyperdense embolic particles within the contralateral prostatic parenchyma, which had just undergone embolization (asterisk) (**b**).

**Table 1 jcm-14-03775-t001:** Baseline data and mean changes of parameters at 12 months.

	1	2	3	4	5	6	7	8	9	10	PAE n = 10
Age, median	74	87	76	78	79	77	81	80	85	83	79.5
Mean IPSS at different time intervals (Pre-PAE—12 mo)	22–10	26–13	23–11	25–12	24–12	24–11	23–13	22–10	26–14	26–11	24–12
Mean maximum urinary flow rate (mL/s) (Pre-PAE—12 mo)	8.1–11.9	7.8–11.6	8.9–12.3	9.2–13.1	8.3–12.2	9.0–12.7	8.6–12.4	8.2–12.0	7.5–11.4	7.7–11.8	8.7–12.6
Mean of prostate volume (cc) (Pre-PAE—12 mo)	66.4–51.5	78.4–53.2	62.1–47.2	54.2–45.6	58.5–46.0	69.9–52.0	65.2–48.2	48.9–43.5	75.3–49.1	72.8–49.8	66.4–49.4
Mean of PSA (ng/mL) (Pre-PAE—12 mo)	5.1–3.5	6.8–4.2	4.6–3.1	3.3–2.5	5.8–3.9	2.9–2.1	5.5–3.6	4.2–3.0	7.2–4.7	6.5–4.0	5.0–3.4
Mean procedure time (min)	92	105	87	73	118	95	81	99	68	110	95
Length of stay (day)	2	3	2	4	3	2	2	2	4	3	2.5

**Table 2 jcm-14-03775-t002:** Study parameters.

Author	Type of Study	No	Age, Mean	Reference
Pisco et al.	Retrospective	630	65.1	[7]
Carnevale et al.	Prospective RCT	30	62	[23]
Amouyal et al.	Retrospective	32	65	[24]
Bagla et al.	Retrospective	78	64.7	[25]
Kurbatov et al.	Prospective	88	66.38	[26]
Grosso et al.	Prospective	13	75.9	[27]
Somani et al.	Prospective	35	64	[28]
Wang et al.	Prospective	115	71.5	[29]
de Assis et al.	Prospective	35	64.8	[30]
Rampoldi et al.	Prospective	43	77.9	[31]
Gonçalves et al.	Prospective	30	NA	[32]
Gabr et al.	Prospective	22	72.5	[33]
Shaker et al.	Prospective	28	68.5	[34]
Gao et al.	Prospective RCT (comparison with TURP)	57	67.7	[35]
Li et al.	Prospective	24	74.5	[36]
Tapping et al.	Prospective	50	67	[37]
Insausti et al.	RCT	45	72.1	[38]
Insausti et al.	Prospective	81	73.87	[39]
Abt et al.	Randomized, open label, and non-inferiority trial	103	65.9	[40]

## Data Availability

The original contributions presented in the study are included in the article, further inquiries can be directed to the corresponding authors.

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
