# Peer review of "Prostate Artery Embolization (PAE) in the Treatment of Benign Prostatic Hyperplasia: A Case Series and Narrative Review"

_jcm, 2025, doi:10.3390/jcm14113775_

Round 1

Reviewer 1 Report

Comments and Suggestions for Authors

Dear Authors,

The manuscript presents data on only ten patients treated by prostate artery embolization (PAE) for BPH; the results are scarce. Benign prostatic hyperplasia (BPH) is the primary cause of male lower urinary tract symptoms (LUTS) and affects more than 50% of men over 60 years old. Recently, prostate artery embolization (PAE) has emerged as an alternative treatment option for LUTS, falling between medical management and surgical options in the spectrum of BPH therapy. Prostatic artery embolization (PAE) is a minimally invasive approach consisting of the occlusion of the prostatic arteries performed under fluoroscopic guidance by trained interventional radiologists. Patient selection for PAE requires unique considerations and meaningful collaboration between urologists and interventional radiologists (IRs) to provide exceptional patient-centric care. 

The evidence supporting PAE now includes over 20 prospective studies and six randomized controlled trials (RCTs): 4 vs. transurethral resection of the prostate (TURP), one vs. sham, and one vs. pharmacotherapy. The longest follow-up includes 2 years post-RCT and 10 years with large cohorts from high-volume centers. (Mouli S, Salem R, McClure TD. Prostate Artery Embolization for Benign Prostatic Hyperplasia. Journal of Urology [Internet]. 2024 Jul 1;212(1):216–9. Available from: https://doi.org/10.1097/JU.0000000000003976).

Regarding the PAE, we already have the following systematic reviews:

Jung JH, McCutcheon KAnn, Borofsky M, Young S, Golzarian J, Kim MH, Dahm P, Narayan VM. Prostatic arterial embolization for treating lower urinary tract symptoms in men with benign prostatic hyperplasia. Cochrane Database of Systematic Reviews 2020, Issue 12. Art. No.: CD012867. DOI: 10.1002/14651858.CD012867.pub2.

Jung JH, McCutcheon KAnn, Borofsky M, Young S, Golzarian J, Kim MH, Narayan VM, Dahm P. Prostatic arterial embolization for the treatment of lower urinary tract symptoms in men with benign prostatic hyperplasia. Cochrane Database of Systematic Reviews 2022, Issue 3. Art. No.: CD012867. DOI: 10.1002/14651858.CD012867.pub3.

and this trial 

  1. Prostatic artery embolisation versus medical treatment in patients with benign prostatic hyperplasia (PARTEM): a randomised, multicentre, open-label, phase 3, superiority trial

    Sapoval, MarcPellerin, Olivier et al. The Lancet Regional Health – Europe, Volume 31, 100672

Author Response

Reviewer Comment:
The manuscript presents data on only ten patients treated by prostate artery embolization (PAE) for BPH; the results are scarce. Benign prostatic hyperplasia (BPH) is the primary cause of male lower urinary tract symptoms (LUTS) and affects more than 50% of men over 60 years old. Recently, prostate artery embolization (PAE) has emerged as an alternative treatment option for LUTS, falling between medical management and surgical options in the spectrum of BPH therapy. Prostatic artery embolization (PAE) is a minimally invasive approach consisting of the occlusion of the prostatic arteries performed under fluoroscopic guidance by trained interventional radiologists. Patient selection for PAE requires unique considerations and meaningful collaboration between urologists and interventional radiologists (IRs) to provide exceptional patient-centric care. The evidence supporting PAE now includes over 20 prospective studies and six randomized controlled trials (RCTs): 4 vs. transurethral resection of the prostate (TURP), one vs. sham, and one vs. pharmacotherapy.

Author Response:
We thank the reviewer for this thoughtful and well-argued comment. We fully acknowledge that the current literature on prostate artery embolization (PAE) is robust and supported by several high-quality randomized controlled trials and systematic reviews, as correctly referenced.

However, the present manuscript was not designed to provide additional evidence of efficacy per se, but rather to serve as a descriptive technical report and documentation of the initial clinical experience from a low-volume center approaching PAE for the first time. This intention has now been explicitly clarified in both the Introduction and Discussion sections. In particular, we have revised the Introduction to state that the primary goal of the study is to offer a detailed account of the technique used, along with preliminary safety and outcome data, which may be of practical value to other institutions planning to implement PAE.

Likewise, the Discussion now opens by clearly framing the study as a technical and procedural report, emphasizing its value in terms of feasibility, patient selection, and early outcomes in a real-world setting. We also revised the Conclusion to reinforce this narrative, stressing the practical implications of the described technique rather than suggesting generalizable efficacy conclusions.

Finally, we have integrated the key references cited in the comment

We hope that this framing helps to better position the manuscript within the existing literature and clarify its intended contribution.

Reviewer 2 Report

Comments and Suggestions for Authors
  • Prostatic artery embolization (PAE) has emerged as a minimally invasive alternative for treating BPH, particularly in high-risk surgical candidates. This study aims to evaluate the efficacy, safety, and clinical outcomes of PAE, but there have been reported before.
  • The numbers of patients are small,so more cases are necessary in the future.
  • Twopatients (20%) underwent unilateral PAE due to vascular calcifications or the presence of an aortobifemoral bypass,what is the indications in unilateral PAE?
  • Which method do you used in this cohort evaluation mean of Prostate volume (cc)in subsequent follow-up assessments after PAE?

Author Response

Reviewer Comment:
Prostatic artery embolization (PAE) has emerged as a minimally invasive alternative for treating BPH, particularly in high-risk surgical candidates. This study aims to evaluate the efficacy, safety, and clinical outcomes of PAE, but there have been reported before. The numbers of patients are small, so more cases are necessary in the future. Two patients (20%) underwent unilateral PAE due to vascular calcifications or the presence of an aortobifemoral bypass. What is the indication in unilateral PAE? Which method did you use in this cohort to evaluate prostate volume (cc) in subsequent follow-up assessments after PAE?

Author Response:
We thank the reviewer for raising these important points. We fully acknowledge the limited number of patients included in our series and agree that larger cohorts are needed in future studies to strengthen the clinical evidence. As clarified in our revised manuscript, the primary objective of this work is not to provide definitive efficacy data but rather to describe our initial technical experience and report early real-world outcomes, which may support other centers in the process of adopting PAE.

Regarding unilateral PAE, the two cases reported in our series were the result of specific anatomical and clinical considerations:

  • In one case, severe prostatic artery calcification prevented safe catheterization of the vessel despite multiple attempts; therefore, the procedure was completed unilaterally after careful intra-procedural evaluation and in agreement with the interventional radiologist.
  • In the second case, the patient had a history of aortobifemoral bypass, which posed a high procedural risk for bilateral embolization. After multidisciplinary discussion between the urology and interventional radiology teams, it was decided to proceed with a unilateral approach as a reasonable compromise between safety and therapeutic benefit.

Regarding the assessment of prostate volume (PV) at follow-up, we confirm that all measurements were consistently performed using transrectal ultrasound (TRUS) at baseline and during each follow-up time point. This method was chosen for its reproducibility, availability, and widespread use in urological practice.

We have now added these clarifications to the revised version of the manuscript to address the reviewer’s concerns.

Reviewer 3 Report

Comments and Suggestions for Authors

This article studies the treatment of BPH with PAE, which has a certain degree of innovation, but it has not been compared with the TURP gold standard.

This may be due to the small sample size. Since the sample size is small, it is recommended that Table 1 present the complete data of 10 patients without the need for an average value!

Author Response

Reviewer Comment:
This article studies the treatment of BPH with PAE, which has a certain degree of innovation, but it has not been compared with the TURP gold standard. This may be due to the small sample size. Since the sample size is small, it is recommended that Table 1 present the complete data of 10 patients without the need for an average value!

Author Response:
We thank the reviewer for the constructive comment. As suggested, we have revised Table 1 to present the complete individual data for all 10 patients, omitting average values. We agree that, given the small sample size, this format provides greater transparency and clarity.

Round 2

Reviewer 1 Report

Comments and Suggestions for Authors

Dear Authors,

The manuscript presents data on only ten patients treated by prostate artery embolization (PAE) for BPH, and the results are scarce. The narrative review of PAE does not offer anything new. 

Reviewer 2 Report

Comments and Suggestions for Authors

The limited number of patients included in the paper. As clarified in the revised manuscript, the primary objective of this work is not to provide definitive efficacy data but rather to describe our initial technical experience and report early real-world outcomes, which may support other centers in the process of adopting PAE.as we know,the author has revised and explained something.